# Exploring factors contributing to antibiotic resistance: A cross-sectional empirical study in Bangladesh

Abdullah Al Rakib[1], Dr. Johaira Sultana Toma[2], Dr. Mousumi Akhtar[3], Md. Abu Hasnat[1,4], Md. Sohel Rana[1]*, Rezwan Ul Haque Aubhi[1], Nur-A.-Alam Mishad[1], Farzana Rahman[1], Shadia Sharmin[1]*

1 Department of Business Administration, University of Scholars, Dhaka, Bangladesh, 2 Medical Officer at Islami Bank Hospital, Dhaka, Bangladesh, 3 FCPS-I, Obstetrics and Gynecology, Bangladesh College of Physicians and Surgeons, Dhaka, Bangladesh, 4 Department of Business Administration, University of Scholars, and Director and Managing Partner, HHH Research Consultancy & Development, Dhaka, Bangladesh

* shadia.sharmin@ius.edu.bd, shadiasharmin2013@gmail.com (SS); sohel@ius.edu.bd (MSR)

## Abstract

Antibiotic resistance is a growing public health concern, particularly in low- and middle-income countries such as Bangladesh. This cross-sectional empirical study analyzed primary survey data collected from 254 respondents in Bangladesh using Structural Equation Modeling (SEM). Structural Equation Modeling (SEM) with Smart-PLS 4.0 was employed to analyze the relationships between key variables, ensuring accurate results despite non-normal data. The findings reveal that economic pressures, inadequate diagnostics, and cultural misconceptions are key contributors to antibiotic overuse and misuse. The study is grounded in the Health Belief Model and Ecological Systems Theory, offering a framework to understand these behaviors. It underscores the need for targeted policies, improved diagnostic resources, and heightened public awareness to address the growing problem of antimicrobial resistance. The findings indicate that economic incentives, diagnostic uncertainty, healthcare infrastructure, and sociocultural beliefs significantly influence antibiotic resistance within the surveyed population.

## 1. Introduction

Antibiotic resistance is one of the most critical global health threats of the 21st century, undermining the effectiveness of treatments and increasing the economic burden on healthcare systems worldwide [1]. According to the World Health Organization (WHO), approximately 1.27 million deaths annually are directly caused by drug-resistant infections, with the brunt of the burden falling on low- and middle-income countries (LMICs) [2]. Regions such as South Asia, including Bangladesh, face unique challenges due to weak healthcare infrastructures, lack of diagnostic facilities, and deeply

**Data availability statement:** The data underlying the results presented in the study are available from: https://doi.org/10.5281/zenodo.19639156.

**Funding:** The author(s) received no specific funding for this work.

**Competing interests:** The authors have declared that no competing interests exist.

rooted socio-cultural practices that exacerbate the misuse of antibiotics [3]. Despite global efforts to mitigate antimicrobial resistance, it is emerging as a silent pandemic, with projections indicating that drug-resistant infections could contribute to 10 million deaths annually by 2050 if no effective action is taken [4]. Beyond mortality, antimicrobial resistance threatens to impose economic costs comparable to the Great Recession of 2008 and could push 24 million people into extreme poverty by 2030 [2].

The drivers of antibiotic resistance are multifaceted, involving economic, diagnostic, systemic, and sociocultural dimensions. Economic incentives often influence physicians' prescribing patterns, as profit-driven motives and patient expectations can lead to overprescription [5]. Diagnostic uncertainty, caused by limited access to accurate and timely tests, further encourages precautionary prescribing [6]. In LMICs, inadequate healthcare infrastructure and diagnostic tools result in empirical use of broad-spectrum antibiotics, accelerating resistance [7,8]. Cultural beliefs and health-seeking behaviors also shape misuse; antibiotics are often viewed as a "quick fix" in many communities, pressuring healthcare providers to prescribe them unnecessarily [9–11]. These complex drivers interact with the evolutionary adaptation of bacterial populations, which develop resistance through genetic modifications or phenotypic tolerance, thereby reducing treatment efficacy [8,12].

Several theoretical perspectives help explain these behaviors and systemic patterns. The Theory of Planned Behavior (TPB) emphasizes the role of attitudes, subjective norms, and perceived behavioral control in shaping antibiotic use [13]. The Health Belief Model (HBM) addresses individual perceptions of risk, severity, and barriers in adopting rational antibiotic practices [14]. Ecological Systems Theory situates these choices within broader healthcare and societal structures, highlighting how systemic limitations reinforce individual and community behaviors [15]. Principal–Agent Theory also provides insight into how misaligned incentives between physicians and patients or healthcare payers can encourage overuse [16].

While a substantial body of literature has examined individual factors in isolation, there is limited research that integrates economic, diagnostic, systemic, and cultural determinants within a unified framework, particularly in the context of Bangladesh. This gap is critical given the country's high burden of antibiotic misuse, constrained diagnostic capacity, and persistent sociocultural drivers [17,18]. The present study addresses this gap by examining the interrelated effects of economic incentives, diagnostic uncertainty, duration of antibiotic use, healthcare infrastructure, and cultural beliefs on antibiotic resistance in Bangladesh. Grounded in the TPB, HBM, and Ecological Systems Theory, the research employs Structural Equation Modeling (SEM) to test hypotheses linking these determinants to increasing antibiotic resistance. By integrating behavioral, economic, and systemic dimensions, this study aims to inform targeted interventions and policy strategies that can mitigate resistance in low-resource settings.

This study is an original empirical research article based on primary quantitative data collected through a structured survey in Bangladesh. Unlike a narrative or systematic review, the study empirically tests theoretically grounded hypotheses using Structural Equation Modeling (SEM) to examine the relationships between economic incentives, healthcare system factors, sociocultural perceptions, and antibiotic

resistance. The manuscript is structured to reflect this empirical focus, with distinct sections for hypothesis development, methodology, results, and discussion of findings.

## 2. Theoretical underpinning and determinants of antibiotic resistance

### 2.1. Theoretical Underpinning

Antibiotic resistance is a global public health crisis shaped by behavioural, economic, and systemic forces. Anchoring this study in established theory enables a rigorous analysis of antibiotic misuse. The Theory of Planned Behaviour (TPB), developed by Icek Ajzen [13], explains how attitudes, subjective norms, and perceived behavioural control shape behaviour. It is particularly relevant for examining cultural beliefs and health-seeking practices, such as societal norms that promote antibiotics as quick remedies for minor illnesses [11,19,20].

The Health Belief Model (HBM), introduced by Irwin M. Rosenstock [14], focuses on perceptions of risk and benefit, highlighting how perceived severity, susceptibility, and barriers influence antibiotic use [14]. Together, TPB and HBM clarify how individual choices interact with social norms to drive misuse.

Economic theory further explains prescribing behaviour. The principal–agent theory shows how misaligned incentives between physicians and patients or payers can lead to overprescription, particularly in profit-driven healthcare systems [16]. Ecological systems theory, developed by Bronfenbrenner [15], situates these behaviours within broader healthcare and societal structures, emphasizing how systemic constraints—such as inadequate diagnostics—encourage empirical and inappropriate antibiotic use.

Integrating these behavioural, economic, and systemic frameworks provides a comprehensive lens for identifying actionable strategies to reduce antibiotic misuse and address this global health challenge.

### 2.2. Determinants of antibiotic resistance and their impact

**2.2.1. Economic incentives impact antibiotic resistance.** Economic incentives significantly influence prescription patterns. Fee-for-service models and profit-driven healthcare systems often encourage overprescription, as physicians may prescribe antibiotics to satisfy patient expectations, avoid conflict, or increase revenue [21]. Pharmaceutical marketing and financial incentives from drug companies further exacerbate this problem [22], undermining rational prescribing and accelerating resistance. Okubo et al. [23] demonstrated that reducing financial incentives through policy interventions led to a significant decline in inappropriate antibiotic prescriptions.

Despite global recognition of the problem, funding for antibiotic stewardship programs remains inadequate, particularly in low-resource settings [24]. Limited financial support and weak enforcement mechanisms hinder their effectiveness. Addressing these challenges requires systemic reform, including realigning healthcare payment structures and reducing pharmaceutical influence on prescribing practices

**2.2.2. Diagnostic uncertainty impacts antibiotic resistance.** Diagnostic uncertainty is a major driver of inappropriate antibiotic use. In resource-limited settings, physicians often rely on empirical treatment due to restricted access to diagnostic tools [17]. Even in well-equipped systems, time pressure can lead to similar overreliance on antibiotics [25]. Improving diagnostic tools and protocols is therefore critical. Llor [26] found that point-of-care (PoC) tests significantly reduced unnecessary prescriptions, while economic models show that PoC diagnostics improve resource efficiency [27].

However, access remains uneven. WHO reports that only a small proportion of primary care facilities in low-income countries have PoC technologies [28]. High costs, weak infrastructure, and insufficient training remain key barriers. Expanding access and training is essential, as PoC implementation has been shown to reduce misuse [29].

**2.2.3. Duration of antibiotic use impacts resistance.** The duration of antibiotic use directly affects resistance. Short courses may fail to eradicate infections, while prolonged use increases selective pressure [30]. Although guidelines

emphasize tailoring duration to specific infections, adherence is inconsistent [31]. Evidence from both a meta-analysis and empirical studies indicates that shorter antibiotic courses can be equally effective for certain infections while reducing the likelihood of resistance development [32,33].

Yet awareness gaps persist. Surveys in 2021 showed that over 40% of healthcare providers in low-resource settings were unaware of updated guidelines [34]. Public and professional awareness campaigns are therefore essential.

**2.2.4. Healthcare Infrastructure and Diagnostic Tools Impact Resistance.** Healthcare infrastructure and access to diagnostics are foundational for effective stewardship. Resource-limited countries face major challenges due to weak antimicrobial stewardship programs [17]. In many low- and middle-income countries, inadequate laboratories and lack of rapid tests force reliance on empirical broad-spectrum antibiotics [35,36]. Previous studies have demonstrated that countries with stronger healthcare infrastructure and better access to diagnostic tools tend to report lower levels of antibiotic resistance [12,37].

Despite awareness, investment gaps persist. In 2023, only 5% of healthcare budgets in low-income countries targeted diagnostics and laboratories [38]. Strengthening healthcare systems and integrating advanced diagnostics are essential [39].

**2.2.5. Cultural beliefs and health-seeking behavior impact resistance.** Cultural beliefs strongly shape antibiotic use. In many societies, antibiotics are perceived as universal remedies, encouraging self-medication and overuse [40]. Physicians also face societal pressure to prescribe, even for viral infections. Educational interventions tailored to cultural contexts reduce misuse by shifting perceptions [41], research in rural India has shown that both community engagement initiatives and the practices of informal healthcare providers play a significant role in shaping antibiotic use and misuse [19,20].

Nevertheless, misconceptions remain widespread. In Bangladesh, 63% of parents believed antibiotics could treat fever and colds [42].

**2.2.6. Increasing antibiotic resistance.** The dependent variable, increasing antibiotic resistance, is a multifaceted issue arising from economic, diagnostic, infrastructural, and cultural factors [43]. Resistance complicates treatment and imposes significant economic and social burdens, with WHO data highlighting the alarming rise in resistance to common antibiotics [44]. Drivers vary across regions: in high-income countries, profit-driven healthcare models and patient expectations often drive overprescription [46], addressed by interventions such as the NHS subscription model [45]. The burden of resistance among key pathogens remains a critical global concern [50].

In low- and middle-income countries, limited access to diagnostic tools and reliance on empirical treatment due to diagnostic uncertainty exacerbate misuse; in Hubei, China, primary care physicians often prescribe antibiotics unnecessarily because of constrained diagnostic capacity [47]. Cultural beliefs further shape antibiotic use: Western societies emphasize immediate solutions [48], whereas many Asian and African contexts expect quick efficacy for minor ailments. Infrastructure gaps in low-income countries, including insufficient healthcare facilities and diagnostic technologies, amplify inappropriate prescribing. Additionally, low prices and sales volume in the antibiotic market hinder investment in new drug development [27]. These drivers differ in significance across contexts, highlighting the need for region-specific strategies that consider local healthcare systems, cultural attitudes, and economic structures. While the root causes of antibiotic resistance are globally similar, targeted interventions must address the context-specific factors driving misuse to effectively mitigate resistance.

## 2.3. Conceptual framework

Fig 1 illustrates the visible issue of increasing antibiotic resistance while highlighting its underlying drivers: economic incentives, diagnostic uncertainty, duration of use, healthcare infrastructure, and cultural beliefs. These deeper factors, often less apparent, play a crucial role in the overuse and misuse of antibiotics, ultimately accelerating resistance.

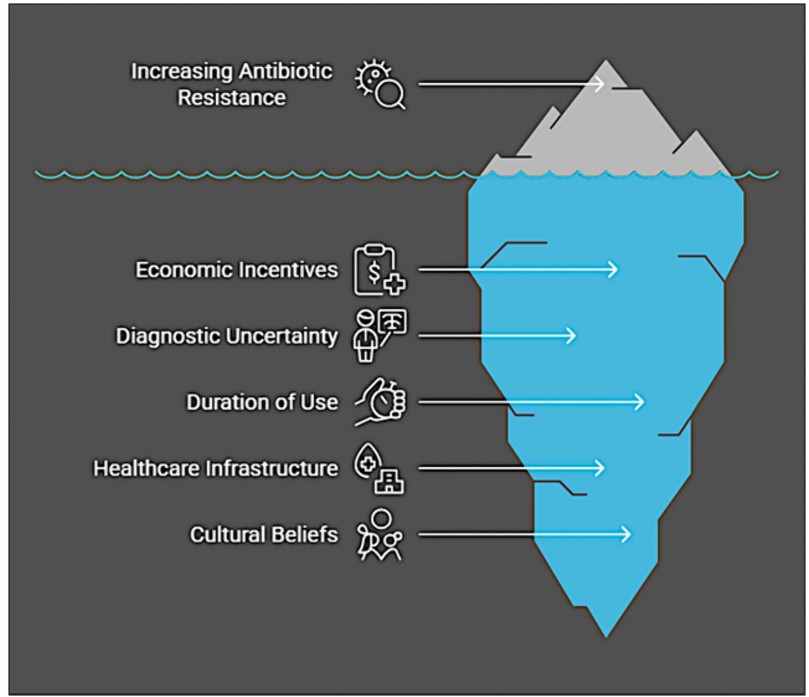

**Fig 1. Factors Contributing to Antibiotic Resistance (Source: Authors' own work).**

Guided by the Theory of Planned Behaviour, the Health Belief Model, principal–agent theory, and ecological systems theory, the conceptual framework integrates behavioural, economic, and structural determinants of antibiotic misuse. Each latent construct in the SEM is theoretically grounded: economic incentives and diagnostic uncertainty capture perceived behavioural control; cultural beliefs reflect subjective norms; treatment duration reflects risk and benefit perceptions; and healthcare infrastructure represents systemic constraints. These theoretically derived pathways inform the development of Hypotheses H1–H5, linking individual and institutional behaviours to increasing antibiotic resistance.

## 3. Hypotheses and operational variables

### 3.1. Hypotheses development

**#H1:** *Economic Incentives Positively Impact Antibiotic Resistance*
   **#H2:** *Diagnostic Uncertainty Significantly and Positively Impacts Antibiotic Resistance*
   **#H3:** *Duration of Antibiotic Use Positively Impacts Antibiotic Resistance*
   **#H4:** *Healthcare Infrastructure Positively Impacts Antibiotic Resistance*
   **#H5:** *Cultural Beliefs and Health-Seeking Behavior Positively Impact Antibiotic Resistance*

### 3.2. Definition of the variables

Table 1 defines six key factors influencing antibiotic resistance: economic incentives, diagnostic uncertainty, treatment duration, healthcare infrastructure, cultural beliefs, and antibiotic misuse. Each variable is supported by literature and measured through survey items, reflecting perceptions on prescribing behaviors, healthcare resources, and regulatory measures.

**Table 1. Definition of operational variables and measurements.**

| Variables | Code | Items | Sources |
|---|---|---|---|
| **Economic Incentives for Doctors:** Economic incentives for doctors refer to financial motivations that can influence prescribing practices, often resulting in the over prescription of antibiotics. This dynamic can lead to an increase in antibiotic resistance as financial rewards may overshadow concerns about the long-term health consequences of unnecessary antibiotic use [49]. | EID1 | Financial incentives make doctors overlook long-term antibiotic resistance risks. | [50] |
| | EID2 | Without incentives, doctors would prescribe antibiotics more cautiously for common illnesses. | |
| | EID3 | Without financial incentives, doctors would prefer alternatives to antibiotics. | |
| **Diagnostic Uncertainty:** Diagnostic uncertainty refers to the lack of clarity or confidence in determining the correct diagnosis for a patient's condition, which can lead healthcare providers to prescribe antibiotics more liberally. This over prescription due to uncertainty can significantly contribute to the rise of antibiotic resistance as inappropriate use becomes more common [51]. | DU1 | Diagnostic uncertainty leads to precautionary antibiotic overuse. | [52] |
| | DU2 | Better diagnosis reduces unnecessary antibiotic prescriptions. | |
| | DU3 | Better diagnostic training lowers antibiotic overprescription. | |
| **Duration of Antibiotic Use:** Duration of antibiotic use pertains to the length of time a patient is prescribed antibiotics for treatment. Extended treatment durations can increase the likelihood of antibiotic resistance, as prolonged exposure to antibiotics can exert selective pressure on bacteria, leading to the emergence of resistant strains [53]. | DAU1 | Monitoring resistant bacteria in trials improves antibiotic policies. | [54]. |
| | DAU2 | Longer treatments encourage antibiotic resistance. | |
| | DAU3 | Not finishing antibiotics increases resistance. | |
| **Healthcare Infrastructure and Diagnostic Tools:** Healthcare infrastructure and diagnostic tools encompass the systems and resources available within healthcare settings that facilitate effective diagnosis and treatment. Improved healthcare infrastructure and access to rapid diagnostic tools can enhance appropriate antibiotic prescribing practices, ultimately reducing the rates of antibiotic resistance [55]. | HIDT1 | Lack of diagnostic tools leads to antibiotic overuse. | [56]. |
| | HIDT2 | Resistance diagnostics improve antibiotic prescriptions. | |
| | HIDT3 | Point-of-care resistance diagnostics (POC-RD) reduces antibiotic misuse and resistance. | |
| **Cultural Beliefs and Health-Seeking Behavior:** Cultural beliefs and health-seeking behavior refer to the societal norms and individual practices related to health and illness, which influence how individuals seek and use healthcare services. These cultural factors can significantly affect antibiotic use and misuse, thereby contributing to the issue of antibiotic resistance when beliefs promote inappropriate access or expectations regarding antibiotic treatment [57]. | CBHSB1 | The "cure-all" belief accelerates antibiotic resistance. | [58] |
| | CBHSB2 | Buying antibiotics without prescriptions fuels resistance. | |
| | CBHSB3 | Correcting misconceptions about antibiotics can slow resistance. | |
| **Increasing Antibiotic Resistance:** Antibiotic resistance is the ability of bacteria to survive and grow despite being exposed to antibiotics, making infections harder to treat and posing a significant global health risk. It is primarily driven by the overuse and misuse of antibiotics in healthcare and agriculture [59]. | IAR1 | Prescribing without tests raises drug resistance risks. | [60] |
| | IAR2 | Stricter OTC antibiotic rules reduce resistance. | |
| | IAR3 | Antibiotics in livestock spread resistance to humans. | |
| | IAR4 | Hospital infection control prevents resistance spread. | |
| | IAR5 | Alternative therapies should be prioritized over antibiotics. | |

(***Source:*** *Authors' own work*)

## 3.3. Scale adaptation and content validation

All measurement items used in this study were adapted from previously validated scales reported in the literature (see Table 1). To ensure their suitability for the Bangladeshi context, a systematic scale adaptation and validation

procedure was followed. First, the original English items were translated into Bangla using a forward–backward translation method. Two bilingual experts independently translated the items into Bangla, and a separate bilingual translator back-translated them into English to ensure semantic equivalence. Any discrepancies were discussed and resolved jointly. Second, the preliminary questionnaire was reviewed by a panel of academic experts with expertise in the study domain and familiarity with the local context. This step ensured content validity, cultural relevance, and clarity of wording. Minor modifications were made based on their feedback to improve comprehensibility and contextual appropriateness. Third, the revised instrument was pre-tested through a pilot survey with a small group of respondents drawn from the target population. Feedback from the pilot test was used to refine item wording and response clarity. The final version of the questionnaire demonstrated satisfactory internal consistency and construct validity, which were further confirmed through reliability and validity analyses reported in the Results section.

## 4. Methodology

This study employs an original empirical, cross-sectional quantitative research design based on primary data collected through a structured survey. The empirical relationships among the study constructs were tested using Structural Equation Modeling (SEM), enabling simultaneous examination of multiple latent variables and hypothesized paths.

### 4.1. Ethics statement

The study was approved by the Research Committee of the concerned university in Bangladesh *(Ethics Number: IUS/Regi.Office/Letter/2024/233).* Informed consent was obtained from all participants before their participation. Consent was explicitly informed, with participants provided clear information about the study objectives, voluntary participation, and their right to skip any question. Consent was collected electronically via Google Forms, where submission of the completed form served as written consent. No personally identifying information was collected to ensure strict anonymity. Since the study did not involve retrospective review of medical records or archived samples, a consent waiver was not applicable.

### 4.2. Sample

The individuals who willingly participated in this research endeavor by completing the survey were provided with a guarantee of the confidentiality of their answers. Employing a convenience sampling technique, an email invitation link through Google Forms was sent to potential respondents. The data collection period was 24/09/24–03/03/25. Subsequently, 254 questionnaires were retrieved, constituting the sample size (N=254).

The study employed a convenience sampling approach using an online questionnaire to collect data from respondents who met predefined eligibility criteria. Participants were included if they were 21 years or older and had experience in the healthcare and related industry. Incomplete questionnaires and responses that did not meet these criteria were excluded from the analysis. Data were collected over a defined period using online recruitment channels, including social media platforms and targeted messaging groups, which allowed efficient access to the relevant population.

A total of 261 responses were initially received, of which 254 valid responses were retained after data screening, resulting in a usable response rate of 97.32%. Prior to analysis, the dataset was examined for missing values and response patterns. The results indicated no substantial missing data, and cases with excessive missing responses were removed following recommended guidelines. The final dataset was then analyzed using SmartPLS (version 4.0) to estimate the measurement and structural models using Partial Least Squares Structural Equation Modeling (PLS-SEM), which is suitable for prediction-oriented research and complex models. This procedure ensured the robustness, transparency, and reproducibility of the empirical analysis.

### 4.3. Measurements and data analysis

The study's questionnaire had seven components: the respondents' demographic characteristics, the scales measuring economic incentives for doctors, diagnostic uncertainty, duration of antibiotic use, healthcare infrastructure and diagnostic tools, cultural beliefs and health-seeking behavior and increasing antibiotic resistance.

SEM was applied through Smart-PLS 4.0 for testing and analysis. This technique is appropriate and suitable for this study as it allows for the simultaneous analysis of multiple dependent relationships and includes latent variables [61]. PLS-SEM has greater statistical power than CB-SEM because of its efficiency in parameter estimation [62]. It is a more reliable analysis method that can be applied to data that does not follow a normal distribution [63].

### 4.4. Scale development, adaptation, and content validation

The measurement items used in this study were adapted from previously validated instruments reported in the literature, as summarized in Table 1. To ensure contextual relevance for the Bangladeshi setting, all items were reviewed for clarity, cultural appropriateness, and consistency with local healthcare practices. The questionnaire was initially prepared in English and subsequently translated into Bangla using a forward–backward translation procedure to preserve semantic equivalence.

Before the main survey, the instrument was pre-tested with a small group of respondents from the target population to assess item clarity, wording, and comprehension. Based on the feedback received, minor modifications were made to improve readability and contextual relevance without altering the underlying constructs. Content validity was further assessed through expert review by researchers with experience in public health and antibiotic use, ensuring that the items adequately captured the intended latent constructs.

The final measurement model was empirically validated using reliability and validity assessments, including internal consistency, convergent validity, and discriminant validity, as reported in the Results section.

## 5. Results

### 5.1. Demographic profile of the respondents

The demographic analysis (Table 2) reveals notable trends within the sample population. The gender distribution reveals a significant predominance of males (65.35%) compared to females (34.65%). In terms of age, the majority of respondents fall within the 31–40 years category (42.52%), followed by the 41–50 years group (22.05%), and the 21–30 years group (18.50%), and above 60 years represent only a small proportion (3.15%). Regarding educational qualifications, a considerable number of respondents are graduates (45.67%), followed by those with higher secondary education (26.77%) and post-graduate degrees (16.93%). Finally, the distribution of years of experience shows that participants with 4–6 years of experience form the largest group (40.94%), followed by those with 0–3 years (31.10%), 7–10 years (20.87%), and more than 10 years of experience constitute 7.09%.

### 5.2. Results of the measurement model

To evaluate the construct reliability and validity, the research examined the measurement model comprehensively, as presented in Table 3. The findings demonstrated robust reliability and internal consistency indicators, with Cronbach's alpha coefficients exceeding the recommended threshold of 0.80, as outlined by [64]. This highlights the constructs' reliability and consistency. The study exhibited methodological rigor by adhering closely to the guidelines established by [65,66].

In addition to Cronbach's alpha, composite reliability (CR) was employed further to assess the internal consistency of the measuring scales. The CR values for the constructs were as follows: 0.946 for economic incentives for doctors, 0.926 for diagnostic uncertainty, 0.764 for the duration of antibiotic use, 0.843 for healthcare infrastructure and diagnostic tools, 0.843 for cultural beliefs and health-seeking behavior, and 0.872 for increasing antibiotic resistance. These findings align with the recommendations of [67,68], emphasizing the reliability and robustness of the constructs.

**Table 2. Sample characteristics.**

| Characteristics | | Frequency | Percent |
|---|---|---|---|
| Gender | Male | 166 | 65.35% |
| | Female | 88 | 34.65% |
| Age | 21-30 | 47 | 18.50% |
| | 31-40 | 108 | 42.52% |
| | 41-50 | 56 | 22.05% |
| | 51-60 | 35 | 13.78% |
| | Above 60 | 08 | 3.15% |
| Education | Secondary | 27 | 16.63% |
| | Higher Secondary | 68 | 26.77% |
| | Graduate | 116 | 45.67% |
| | Post-graduate | 43 | 16.93% |
| Working Experience | 0-3 Years | 79 | 31.10% |
| | 4-6 Years | 104 | 40.94% |
| | 7-10 Years | 53 | 20.87% |
| | Above 10 Years | 18 | 7.09% |

(**Source:** *Customized output of SPSS*)

**Table 3. Convergent Validity.**

| Constructs | Items | Loadings | α | CR | AVE |
|---|---|---|---|---|---|
| Economic Incentives for Doctors | EID1 | 0.835 | 0.902 | 0.946 | 0.835 |
| | EID2 | 0.954 | | | |
| | EID3 | 0.947 | | | |
| Diagnostic Uncertainty | DU1 | 0.876 | 0.919 | 0.926 | 0.861 |
| | DU2 | 0.962 | | | |
| | DU3 | 0.943 | | | |
| Duration of Antibiotic Use | DAU1 | 0.858 | 0.762 | 0.764 | 0.677 |
| | DAU2 | 0.813 | | | |
| | DAU3 | 0.796 | | | |
| Healthcare Infrastructure and Diagnostic Tools | HIDT1 | 0.899 | 0.828 | 0.843 | 0.746 |
| | HIDT2 | 0.906 | | | |
| | HIDT3 | 0.780 | | | |
| Cultural Beliefs and Health-Seeking Behaviour | CBHSB1 | 0.861 | 0.840 | 0.843 | 0.758 |
| | CBHSB2 | 0.888 | | | |
| | CBHSB3 | 0.862 | | | |
| Increasing Antibiotic Resistance | IAR1 | 0.883 | 0.862 | 0.872 | 0.645 |
| | IAR2 | 0.785 | | | |
| | IAR3 | 0.788 | | | |
| | IAR4 | 0.743 | | | |
| | IAR5 | 0.809 | | | |

*Abbreviations: CR = Composite Reliability, AVE = Average Variance Extracted, α = Cronbach's alpha. ***All indicators are significant at p < 0.001 (**Source:** Customized output of Smart-PLS)*

The study also emphasized the importance of Average Variance Extracted (AVE) in measuring convergent validity. As described by [69] and supported by [70], AVE quantifies the variance explained by a construct relative to the variance attributed to measurement error. To ensure strong convergent validity, achieving an AVE of at least 0.50 is critical, as recommended by [71,72]. An AVE below 0.50 would suggest that measurement errors outweigh the variance explained by the construct, underscoring the need to meet this threshold for validity. In this study, all constructs exceeded the minimum AVE requirement of 0.50, with AVE values ranging above 0.645. This compliance demonstrates the rigorous validation process, ensuring the constructs' reliability and validity. By adhering to the methodological recommendations of [69,71,73], the research showcases a robust framework for evaluating construct validity and reliability.

According to Rasoolimanesh [74], emphasize that discriminant validity is a critical aspect of PLS-SEM path analysis, ensuring the statistical distinction between latent variables representing separate theoretical constructs. The results presented in Table 4 confirm the achievement of discriminant validity by satisfying the rigorous criteria of the Heterotrait-Monotrait Ratio (HTMT). According to Hair and Henseler [75,76], the HTMT metric is a reliable tool for assessing the degree of similarity between two latent variables. For discriminant validity to be established, HTMT values must fall below the threshold of 1. This study's findings align with this requirement and demonstrate compliance with the methodological standards, providing robust evidence of discriminant validity.

## 5.3. Goodness of fit

The model's effectiveness and fit were evaluated using the Coefficient of Determination ($R^2$), the Standardized Root Mean Squared Residual (SRMR), and the Normed Fit Index (NFI). According to Janadari [77], an $R^2$ value between 0.25 and 0.50 is considered optimal, with any value above 0.20 deemed acceptable. Furthermore, Sharma et al. [78] emphasize that an $R^2$ value closer to 1 indicates more substantial explanatory power, with 0.25 serving as the threshold for meaningful effect sizes in path models. The present study's $R^2$ value of 0.882 demonstrates a highly significant relationship between variables, as shown in Table 5. Moreover, Plonsky and Ghanba [79] highlight the importance of adjusted $R^2$, which accounts for the number of independent variables in a regression model. This adjustment enhances the robustness

**Table 4. Discriminant Validity (HTMT Ratio).**

|  | 1 | 2 | 3 | 4 | 5 | 6 |
|---|---|---|---|---|---|---|
| 1. EID |  |  |  |  |  |  |
| 2. DU | 0.728 |  |  |  |  |  |
| 3. DAU | 0.667 | 0.720 |  |  |  |  |
| 4. HIDT | 0.773 | 0.631 | 0.684 |  |  |  |
| 5. CBHSB | 0.707 | 0.777 | 0.756 | 0.727 |  |  |
| 6. IAR | 0.567 | 0.758 | 0.765 | 0.785 | 0.710 |  |

*Abbreviations:* EID = Economic Incentives for Doctors, DU = Diagnostic Uncertainty, DAU = Duration of Antibiotic Use, HIDT = Healthcare Infrastructure and Diagnostic Tools, CBHSB = Cultural Beliefs and Health-Seeking Behavior, and IAR = Increasing Antibiotic Resistance (***Source:*** Smart-PLS output)

**Table 5. Model fitness.**

| Name of Criteria | Value |
|---|---|
| $R^2$ | 0.882 |
| SRMR | 0.062 |
| NFI | 0.822 |

(***Source:*** Smart-PLS output)

of the model, aligning with the comprehensive analytical approach used in this research. Regarding model fit, [80,81] suggest that a model is well-fitted if the NFI value approaches one and the SRMR is below 0.08. The study's findings include an NFI value of 0.822, close to the ideal of 1, and an SRMR value of 0.062, comfortably below the 0.08 threshold. These metrics, as outlined in Table 5, indicate that the model demonstrates an acceptable and reliable fit, consistent with the guidelines provided by [81].

The model demonstrates strong explanatory power, accounting for a high proportion of variance in increasing antibiotic resistance ($R^2 = 0.882$), which, while uncommon in many behavioral and social science studies, is plausible in this context due to the inclusion of theoretically grounded and conceptually proximal predictors capturing interconnected behavioral, institutional, and regulatory drivers. Interpretation of standardized path coefficients and effect sizes indicates that the predictors differ in their practical influence, underscoring the importance of considering effect magnitude alongside statistical significance when evaluating real-world relevance. Nevertheless, this high $R^2$ should be interpreted with caution, as it may partly reflect overlapping constructs, context-specific dynamics within the Bangladeshi healthcare and regulatory environment, and the cross-sectional nature of the data, which limits causal inference. Although model diagnostics suggest acceptable fit and no critical multicollinearity, the possibility of context-bound overfitting cannot be entirely excluded. Accordingly, the findings should be viewed as demonstrating strong explanatory capacity within the studied setting rather than universal predictive validity, highlighting the need for replication across different contexts and the use of longitudinal or external validation approaches in future research.

## 5.4. Hypothesis testing

In Table 6 and Fig 2, the first hypothesis (H1) explores the impact of economic incentives for doctors on antibiotic resistance. This relationship is strongly supported by a high T-statistic of 6.501 and a p-value of 0.000. The second hypothesis (H2) addresses diagnostic uncertainty as a factor. The results show a T-statistic of 4.271 and a p-value of 0.001, indicating a statistically significant relationship. The third hypothesis (H3) examines the duration of antibiotic use and its influence. The T-statistic of 5.759 and p-value of 0.001 strongly support this hypothesis, emphasizing that prolonged or inappropriate antibiotic usage is a major contributor to resistance development. Hypothesis four (H4) investigates the role of healthcare infrastructure and diagnostic tools. The results confirm a significant association with a T-statistic of 3.213 and a p-value of 0.011. Finally, hypothesis five (H5) explores the influence of cultural beliefs and health-seeking behaviour. Although this relationship is the least statistically significant among the five, with a T-statistic of 2.083 and a p-value of 0.021, it is still supported.

## 6. Discussion of findings

This study provides robust empirical evidence on the multifactorial drivers of antibiotic resistance by integrating economic, diagnostic, systemic, and sociocultural determinants within a unified Structural Equation Modeling framework. All five

**Table 6. Result of hypothesis testing.**

| Hypothesis | Relationships | T statistics | p-values | Decision |
|---|---|---|---|---|
| H1 | EID -> IAR | 6.501 | 0.000 | Supported |
| H2 | DU -> IAR | 4.271 | 0.001 | Supported |
| H3 | DAU -> IAR | 5.759 | 0.001 | Supported |
| H4 | HIDT -> IAR | 3.213 | 0.011 | Supported |
| H5 | CBHSB -> IAR | 2.083 | 0.021 | Supported |

*Abbreviations:* EID = Economic Incentives for Doctors, DU = Diagnostic Uncertainty, DAU = Duration of Antibiotic Use, HIDT = Healthcare Infrastructure and Diagnostic Tools, CBHSB = Cultural Beliefs and Health-Seeking Behavior, and IAR = Increasing Antibiotic Resistance (***Source***: Smart-PLS output)

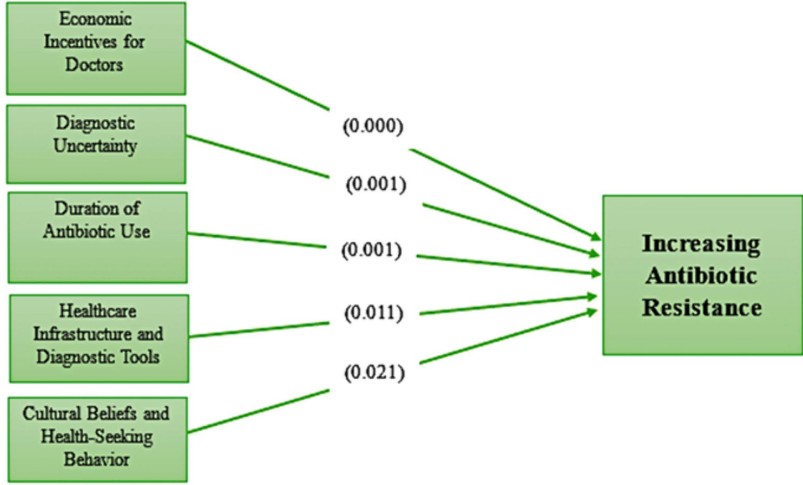

**Fig 2. Results of the path analysis (Source: Authors' own work).**

hypothesized relationships (H1-H5) were statistically supported, underscoring that antibiotic resistance is not the result of a single behavioral failure but rather emerges from interconnected individual, institutional, and societal mechanisms. Interpreting these findings through established theoretical lenses including the Theory of Planned Behavior, the Health Belief Model, Principal-Agent Theory, and Ecological Systems Theory offers important insights into how antibiotic misuse is perpetuated in low- and middle-income countries, particularly Bangladesh [82–84].

### Economic incentives and antibiotic resistance (H1)

The strong and highly significant association between economic incentives and increasing antibiotic resistance highlights the central role of financial motivations in shaping prescribing behavior. In market-oriented and predominantly out-of-pocket healthcare systems such as Bangladesh, physicians often operate under financial pressures that may unintentionally encourage overprescription. This finding is consistent with Principal-Agent Theory, which posits that misaligned incentives between healthcare providers and patients can lead to suboptimal decision-making [85]. When revenue generation, patient satisfaction, or pharmaceutical influence outweighs long-term public health considerations, antibiotics are more likely to be prescribed unnecessarily.

The magnitude of this relationship suggests that economic drivers may exert a stronger influence on antibiotic use in LMICs than in regulated, publicly funded healthcare systems. Importantly, this result reinforces calls for systemic reforms, such as decoupling physician income from prescription volume, strengthening regulatory oversight of pharmaceutical marketing, and expanding antibiotic stewardship programs that address financial conflicts of interest [86].

### Diagnostic uncertainty and antibiotic resistance (H2)

Diagnostic uncertainty emerged as a significant predictor of antibiotic resistance, reflecting the pervasive reliance on empirical prescribing in settings with limited access to reliable diagnostic tools [7]. In Bangladesh, diagnostic facilities are unevenly distributed, and primary healthcare providers frequently lack timely laboratory confirmation, leading to precautionary antibiotic use even when bacterial infection is uncertain. This behavior aligns with risk-avoidance strategies described in the clinical decision-making literature, where antibiotics are prescribed to mitigate potential adverse outcomes under uncertainty. The findings underscore that antibiotic misuse is not solely a consequence of poor clinical judgment but is often a rational response to structural constraints. Strengthening diagnostic capacityparticularly through

affordable point-of-care testing and provider trainingmay therefore yield substantial reductions in inappropriate antibiotic use. This result reinforces global antimicrobial stewardship frameworks that emphasize diagnostics as a cornerstone of resistance mitigation [87].

### Duration of antibiotic use and resistance development (H3)

The statistically strong relationship between the duration of antibiotic use and increasing resistance highlights the critical role of treatment adherence and prescribing practices [10]. Both excessively prolonged antibiotic courses and premature discontinuation contribute to selective pressure on bacterial populations, accelerating resistance development. In the Bangladeshi context, inconsistent adherence often reflects inadequate patient counseling, self-medication practices, and limited follow-up mechanisms within the healthcare system.

This finding reinforces emerging evidence that shorter, evidence-based antibiotic regimens can be equally effective for many infections while reducing resistance risk. The results emphasize the need for clearer clinical guidelines, improved provider-patient communication, and public education initiatives aimed at correcting misconceptions about antibiotic duration.

### Healthcare Infrastructure and Diagnostic Capacity (H4)

Healthcare infrastructure and access to diagnostic tools were also significantly associated with antibiotic resistance, underscoring the systemic nature of the problem [5]. Weak laboratory capacity, shortages of trained personnel, and insufficient integration of antimicrobial stewardship practices compel healthcare providers to rely on broad-spectrum antibiotics as default treatment options [9]. This empirical evidence supports Ecological Systems Theory, which emphasizes that individual behaviors are shaped by broader institutional and policy environments.

The findings suggest that antibiotic resistance in Bangladesh cannot be effectively addressed without substantial investment in healthcare infrastructure. Improving laboratory networks, integrating rapid diagnostics into routine care, and strengthening health system governance are essential steps toward reducing empirical prescribing and mitigating resistance at scale.

### Cultural Beliefs, Health-Seeking Behavior, and Antibiotic Resistance (H5)

Although cultural beliefs and health-seeking behavior exhibited the weakest statistical association among the five predictors, the relationship remained significant and substantively meaningful [88]. In many communities, antibiotics are perceived as a universal remedy, and patients often expect or demand antibiotics for self-limiting conditions such as viral infections [11]. These expectations place pressure on healthcare providers and normalize inappropriate antibiotic use.

The relatively weaker effect size suggests that cultural factors may influence antibiotic resistance indirectly, operating through patient demand, informal healthcare providers, and self-medication practices rather than directly shaping prescribing decisions. This finding aligns with the Health Belief Model, which emphasizes perceived benefits and barriers in health-related decision-making. Addressing these cultural drivers requires sustained, context-sensitive public health education and community engagement rather than isolated awareness campaigns.

### Integrated Interpretation and Broader Implications

Taken together, the findings confirm that antibiotic resistance is a complex, multidimensional phenomenon requiring coordinated interventions across economic, clinical, systemic, and sociocultural domains [82]. The strong explanatory power of the model highlights the value of integrating behavioral theories with health systems perspectives. Importantly, the results support global policy frameworks, such as the WHO Global Action Plan on Antimicrobial Resistance [82], which advocate for a holistic, cross-sectoral approach. From a theoretical standpoint, this study extends existing models by empirically

demonstrating how economic and structural constraints interact with individual beliefs and behaviors to shape antibiotic misuse. From a practical perspective, the findings indicate that isolated interventionssuch as public awareness campaigns aloneare unlikely to succeed unless accompanied by financial reforms, diagnostic investment, and health system strengthening [8]. Addressing antibiotic resistance, therefore, requires not only changing individual behavior but also redesigning the environments in which prescribing and consumption decisions occur.

### 6.1. Theoretical implications

This study extends the application of the Theory of Planned Behavior (TPB) and the Health Belief Model (HBM) by integrating economic constraints and healthcare system inefficiencies as key determinants of antibiotic misuse. While TPB explains how individual attitudes, perceived behavioral control, and subjective norms shape antibiotic consumption, this research highlights how economic hardships and limited healthcare access reduce perceived control, increasing self-medication and misuse. Similarly, HBM's concept of perceived susceptibility is expanded by demonstrating how societal misconceptions and limited awareness downplay the long-term risks of antibiotic resistance. Furthermore, by applying Ecological Systems Theory, the study underscores how micro-level behaviors (individual beliefs and financial constraints) are shaped by macro-level influences (government policies, pharmaceutical regulations, and healthcare infrastructure), necessitating a systemic approach to antibiotic resistance mitigation.

### 6.2. Practical implications

Findings from this research emphasize the need for policy interventions and healthcare reforms to curb antibiotic misuse, particularly in economically disadvantaged populations in an emerging economy like Bangladesh. Strengthening prescription regulations, expanding access to affordable healthcare, and implementing antibiotic stewardship programs can mitigate the negative impact of economic and systemic barriers. Additionally, targeted public health campaigns must address societal misconceptions and increase awareness about antibiotic resistance, leveraging culturally relevant communication strategies to improve adherence to responsible antibiotic use. Collaboration between governments, healthcare providers, and the pharmaceutical industries is essential to create sustainable economic and healthcare policies prioritizing long-term antibiotic efficacy and public health protection.

## 7. Conclusion, limitations, and future research directions

This study provides empirical evidence on the determinants of antibiotic resistance based on primary survey data collected from a convenience sample in Bangladesh. Based on established theoretical frameworks such as the Theory of Planned Behavior, the Health Belief Model, Principal-Agent Theory, and Ecological Systems Theory, this study demonstrates how economic incentives, diagnostic uncertainty, antibiotic duration, healthcare infrastructure, and cultural beliefs contribute to antibiotic misuse. By evaluating these drivers, the study emphasizes the critical need for comprehensive policies to tackle the global health emergency. Tailored treatments aimed at individual behaviors, systemic inefficiencies, and regional inequities are critical for combating the growing issue of antibiotic resistance.

Despite the strengths of this study, several limitations warrant consideration. First, relying on existing literature may restrict the understanding of evolving factors influencing antibiotic resistance in rapidly changing healthcare environments. Second, the cross-sectional research design does not allow for causal inference or examination of changes in antibiotic use behaviors over time. Additionally, cultural beliefs and health-seeking behaviors were explored broadly, but region-specific nuances may not have been fully captured. Lastly, while theoretical frameworks provided a solid foundation, their practical application to policy-making and intervention design was not extensively analyzed, which limits the actionable insights derived from this research. Addressing these limitations and advancing research in these areas can strengthen the fight against antibiotic resistance, paving the way for more effective and equitable healthcare practices.

To build upon the findings of this research, future studies should prioritize longitudinal and region-specific analyses to understand how cultural, economic, and systemic factors evolve. Expanding research to high-income settings can offer comparative insights and highlight universal versus context-specific drivers of resistance. Further exploration into the role of pharmaceutical marketing and informal healthcare providers is necessary to address overlooked contributors to antibiotic misuse. Moreover, interdisciplinary studies incorporating behavioral economics and health systems research can enhance the development of more targeted interventions. Finally, future research should evaluate the effectiveness of antibiotic stewardship programs and innovative diagnostic tools in mitigating resistance, ensuring that interventions are evidence-based and adaptable to diverse healthcare systems. Addressing these limitations in future research can strengthen evidence-based efforts to address antibiotic resistance in Bangladesh and comparable settings.

## Author contributions

**Conceptualization:** Abdullah Al Rakib, Dr. Johaira Sultana Toma, Mousumi Akhtar, Rezwan Ul Haque Aubhi.

**Data curation:** Abdullah Al Rakib, Dr. Johaira Sultana Toma, Mousumi Akhtar.

**Formal analysis:** Abdullah Al Rakib, Md. Sohel Rana, Nur-A-Alam Mishad.

**Investigation:** Abdullah Al Rakib.

**Methodology:** Abdullah Al Rakib, Md. Sohel Rana, Rezwan Ul Haque Aubhi.

**Resources:** shadia sharmin.

**Software:** shadia sharmin.

**Supervision:** Md. Abu Hasnat, Md. Sohel Rana, shadia sharmin.

**Validation:** Dr. Johaira Sultana Toma, Md. Abu Hasnat, Rezwan Ul Haque Aubhi, Farzana Rahman, shadia sharmin.

**Visualization:** Md. Abu Hasnat, Md. Sohel Rana.

**Writing – original draft:** Abdullah Al Rakib, Dr. Johaira Sultana Toma, Mousumi Akhtar, Md. Abu Hasnat, Md. Sohel Rana, Rezwan Ul Haque Aubhi, Nur-A-Alam Mishad, Farzana Rahman.

**Writing – review & editing:** Md. Abu Hasnat, Md. Sohel Rana, shadia sharmin.

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
