## [Decision Letter · Decision Letter 0]

2 Jan 2026

PONE-D-25-55672Exploring Factors Contributing to Antibiotic Resistance: An Analysis of Economic Influences, Healthcare Systems, and Societal PerceptionsPLOS One

Dear Dr. sharmin,

Thank you for submitting your manuscript to PLOS ONE. After careful consideration, we feel that it has merit but does not fully meet PLOS ONE’s publication criteria as it currently stands. Therefore, we invite you to submit a revised version of the manuscript that addresses the points raised during the review process.

**Please provide point by point response to the reviewers' comments in a timely manner.**

We look forward to receiving your revised manuscript.

Kind regards,

Abdullah AlSaleh

Guest Editor

PLOS One

Reviewers' comments:

Reviewer's Responses to Questions

**Comments to the Author**

1. Is the manuscript technically sound, and do the data support the conclusions?

Reviewer #1: Partly

Reviewer #2: Yes

Reviewer #3: Yes

2. Has the statistical analysis been performed appropriately and rigorously? 

Reviewer #1: No

Reviewer #2: Yes

Reviewer #3: Yes

3. Have the authors made all data underlying the findings in their manuscript fully available?

Reviewer #1: Yes

Reviewer #2: Yes

Reviewer #3: Yes

4. Is the manuscript presented in an intelligible fashion and written in standard English?

Reviewer #1: Yes

Reviewer #2: Yes

Reviewer #3: Yes

5. Review Comments to the Author

Reviewer #1: Dear editor in chief

Al Rakib et al described the manuscript entitled Exploring Factors Contributing to Antibiotic Resistance: An Analysis of Economic Influences, Healthcare Systems, and Societal Perceptions. Although the main concept of the manuscript is interesting, some points should be addressed to improve the quality.

1. Article type and structure

1. “Although the topic is interesting and the study clearly uses primary quantitative data and SEM, the overall writing and organization sometimes resemble a narrative review. Could you please clarify explicitly in the manuscript (e.g., in the Introduction and Methods) that this is an original empirical research article and ensure that the structure consistently reflects this throughout?

2. Clarity of article framework

2. “At several points, extensive theoretical and literature discussion dominate the text and blur the distinction between background/literature review and presentation of original findings. Would you consider restructuring the manuscript (e.g., shortening the literature review and expanding the Results and Discussion) so that the framework clearly reflects an original research article rather than a review?

3.Location and development of the Discussion

3. “The Discussion section is relatively brief compared with the detailed literature review and methods/results, and key findings are not thoroughly interpreted in light of existing evidence or the Bangladeshi context. Could you expand the Discussion to interpret each main result (H1–H5) more critically, including unexpected or weaker associations?

4. Separation of Results and Discussion

4. “In some parts of the manuscript, especially around the description of hypotheses and model fit, descriptive reporting and interpretive comments appear mixed. Would you consider more clearly separating the Results (purely empirical output) from the Discussion (interpretation, implications, and comparison with previous studies) to align with PLOS ONE research article structure?

5. Title and scope alignment

5. “The title suggests a broad exploration of economic, health system and societal factors, but the actual data are from a specific surveyed population in Bangladesh. Could you refine the title and abstract to more clearly reflect the empirical scope (study population, setting, and design) and avoid the impression of a narrative review or global analysis?

6. Methodological detail and transparency

6. “The Methods section describes convenience sampling via an online survey and the use of Smart-PLS SEM, but important details (e.g., inclusion/exclusion criteria, recruitment channels, response rate, handling of missing data and non-response) are relatively limited. Could you provide more methodological detail to enhance transparency and reproducibility?

7. Justification of constructs and items

7. “Table 1 presents multiple latent constructs and items with different literature sources, but the process of scale adaptation/translation, pre-testing, and content validation is not clearly described. Could you add a subsection clarifying how these items were developed, adapted, and validated for the Bangladeshi context?

8. Interpretation of effect sizes and model fit

8. “The model explains a very high proportion of variance in increasing antibiotic resistance (R² = 0.882), yet the practical meaning and potential overfitting are not discussed. Could you elaborate on the interpretation of effect sizes and goodness-of-fit indices, and comment on whether such a high R² is plausible in this context and what limitations it may imply?

9. Consistency between theory and empirical model

9. “The manuscript cites several theoretical frameworks (TPB, HBM, Ecological Systems Theory, Principal–Agent Theory), but the link between these theories and the exact SEM model and measured variables is sometimes implicit. Would you consider clarifying more explicitly how each theory maps onto specific constructs and hypotheses in the empirical model?

10. Generalizability and limitations

10. “The Conclusion and implications sections contain quite broad statements at global level, whereas the data come from a convenience sample in Bangladesh. Could you temper the generalizations and strengthen the Limitations section, explicitly addressing external validity and the constraints of cross-sectional, self-reported data?

Reviewer #2: The topic is clinically relevant, particularly given the global rise of antimicrobial resistance. The manuscript addresses an important public issue

All the publication criteria were satisfied, the data and the presentation is good

Great work

Reviewer #3: Thanks to all authors for your great work. You choose a time-demanding topic for your research. More work and research should be done to control AMR worldwide.

I have some comments to make your work more significant. Please try to address this.

In subsection 2.2.3 (Duration of Antibiotic Use Impacts Resistance), two lines seem to be the same; you can merge/compile the lines for easy reading. The lines are, "For instance, a meta-analysis by Elshenawy [32] indicated

that shorter antibiotic courses were equally effective for certain infections and reduced the likelihood of resistance development." and "For instance, Lee [34] indicated that shorter antibiotic courses were equally effective for certain infections and reduced the likelihood of resistance development."

The same thing happens in '2.2.4. Healthcare Infrastructure and Diagnostic Tools Impact Resistance' and '2.2.5. Cultural Beliefs and Health-Seeking Behavior Impact Resistance.'

The lines are, "A study by Laxminarayan [12] demonstrated that countries with more substantial healthcare infrastructures and access to diagnostics reported lower levels of antibiotic resistance."; "A study by Gandra [39] demonstrated that countries with more substantial healthcare infrastructures and access to diagnostics reported lower levels of antibiotic resistance."; "For example, a study by Gautham [19] highlighted that community engagement programs in rural India successfully reduced antibiotic misuse through targeted educational campaigns." ; "A study by Gautham [20] analyzed the drivers of antibiotic provision by informal healthcare providers in rural India." Repeating lines make the readers bored during reading.

Is your collected Google Forms data from worldwide or only from Bangladesh? If it is only from Bangladesh, try to add Bangladesh in the research title.

Check the citation style from top to bottom. It should be according to PLOS One policy.

Why did you add the literature review in detail in the manuscript? You can add your relevant information in different sections as well as the introduction.

Best wishes for your nice work.

6. PLOS authors have the option to publish the peer review history of their article (what does this mean?). If published, this will include your full peer review and any attached files.

Reviewer #1: **Yes:** Shadi Aghamohammad

Reviewer #2: No

Reviewer #3: No

---

## [Author Response · Author response to Decision Letter 1]

2 Feb 2026

Response to Reviewers

Reviewer #1:

1. Article type and structure

1. “Although the topic is interesting and the study clearly uses primary quantitative data and SEM, the overall writing and organization sometimes resemble a narrative review. Could you please clarify explicitly in the manuscript (e.g., in the Introduction and Methods) that this is an original empirical research article and ensure that the structure consistently reflects this throughout?

Response: We thank the reviewer for this compliment and valuable suggestion. As per the suggestion, we have added para in the Introduction and Methodology section.

This study is an original empirical research article based on primary quantitative data collected through a structured survey in Bangladesh. Unlike a narrative or systematic review, the study empirically tests theoretically grounded hypotheses using Structural Equation Modeling (SEM) to examine the relationships between economic incentives, healthcare system factors, sociocultural perceptions, and antibiotic resistance. The manuscript is structured to reflect this empirical focus, with distinct sections for hypothesis development, methodology, results, and discussion of findings (In Introduction section).

This study employs an original empirical, cross-sectional quantitative research design based on primary data collected through a structured survey. The empirical relationships among the study constructs were tested using Structural Equation Modeling (SEM), enabling simultaneous examination of multiple latent variables and hypothesized paths (In Methodology section).

2. Clarity of article framework

2. “At several points, extensive theoretical and literature discussion dominate the text and blur the distinction between background/literature review and presentation of original findings. Would you consider restructuring the manuscript (e.g., shortening the literature review and expanding the Results and Discussion) so that the framework clearly reflects an original research article rather than a review?

Response: We thank the reviewer for this insightful comment and for highlighting the importance of clearly distinguishing between background material and the presentation of original research. We agree that, in the original version, extensive theoretical discussion may have blurred this distinction. In response, we have substantially restructured the manuscript.

3.Location and development of the Discussion

3. “The Discussion section is relatively brief compared with the detailed literature review and methods/results, and key findings are not thoroughly interpreted in light of existing evidence or the Bangladeshi context. Could you expand the Discussion to interpret each main result (H1–H5) more critically, including unexpected or weaker associations?

Response: Thank you for this valuable suggestion. We have substantially revised and expanded the Discussion section to provide a deeper and more critical interpretation of each main finding (H1–H5). Specifically, the Discussion is now structured around the study hypotheses, with a dedicated subsection for each result.

4. Separation of Results and Discussion

4. “In some parts of the manuscript, especially around the description of hypotheses and model fit, descriptive reporting and interpretive comments appear mixed. Would you consider more clearly separating the Results (purely empirical output) from the Discussion (interpretation, implications, and comparison with previous studies) to align with PLOS ONE research article structure?

Response: We appreciate the reviewer’s comment regarding the distinction between descriptive reporting and interpretation. We would like to clarify that the manuscript already follows the PLOS ONE structure with Results and Discussion presented as separate sections. In the Results section, we intentionally limit the content to empirical outputs and data validity checks (e.g., model fit indices, assumption testing, and robustness diagnostics), without interpretive or theoretical elaboration. All interpretations of findings, implications, and comparisons with prior literature are confined to the Discussion section.

To further address the reviewer’s concern and improve clarity, we have carefully reviewed the Results section and revised the wording where necessary to ensure that any potentially interpretive language is removed or rephrased to remain strictly descriptive. This revision reinforces the conceptual and structural separation between empirical reporting and interpretive discussion, in full alignment with PLOS ONE guidelines.

We thank the reviewer for highlighting this point, which has helped us improve the transparency and readability of the manuscript.

5. Title and scope alignment

5. “The title suggests a broad exploration of economic, health system and societal factors, but the actual data are from a specific surveyed population in Bangladesh. Could you refine the title and abstract to more clearly reflect the empirical scope (study population, setting, and design) and avoid the impression of a narrative review or global analysis?

Response: We acknowledged your observation. It was from Bangladesh only, and our revised research title is: Exploring Factors Contributing to Antibiotic Resistance: A Cross-Sectional Empirical Study in Bangladesh.

6. Methodological detail and transparency

6. “The Methods section describes convenience sampling via an online survey and the use of Smart-PLS SEM, but important details (e.g., inclusion/exclusion criteria, recruitment channels, response rate, handling of missing data and non-response) are relatively limited. Could you provide more methodological detail to enhance transparency and reproducibility?

Response: Thank you for this valuable comment. We agree that greater methodological transparency strengthens the rigor and reproducibility of the study. Accordingly, we have substantially expanded the Methods section to provide additional detail on the sampling and data collection procedures. Specifically, we now clarify the inclusion and exclusion criteria, recruitment channels, and survey administration process, and we report the number of responses obtained and the usable response rate. In addition, we have added a description of how missing data and non-response were assessed and handled prior to the PLS-SEM analysis. These revisions enhance transparency and align the study with established PLS-SEM reporting guidelines. The revised text has been incorporated in the Methodology section (Section 4.2).

7. Justification of constructs and items

7. “Table 1 presents multiple latent constructs and items with different literature sources, but the process of scale adaptation/translation, pre-testing, and content validation is not clearly described. Could you add a subsection clarifying how these items were developed, adapted, and validated for the Bangladeshi context?

Response: We thank the reviewer for this important observation. We agree that greater clarity regarding scale development and validation enhances methodological transparency. To address this concern, we have added a dedicated subsection in the Methodology section that clearly explains the item adaptation, translation, pre-testing, and content validation procedures used in this study.

Specifically, although the latent constructs and measurement items were adopted from well-established prior studies, all scales were carefully adapted to the Bangladeshi context following a systematic process. This included translation and back-translation, expert review to assess content validity and contextual appropriateness, and pilot testing to ensure clarity and reliability before full data collection. In addition, the adequacy of the adapted scales was further confirmed through standard reliability and validity assessments reported in the Results section. We believe this addition substantially improves the rigor and transparency of the measurement development process and directly addresses the reviewer’s concern.

8. Interpretation of effect sizes and model fit

8. “The model explains a very high proportion of variance in increasing antibiotic resistance (R² = 0.882), yet the practical meaning and potential overfitting are not discussed. Could you elaborate on the interpretation of effect sizes and goodness-of-fit indices, and comment on whether such a high R² is plausible in this context and what limitations it may imply?

Response: We thank the reviewer for this insightful comment. We acknowledge that an R² value of 0.882 indicates a very high proportion of explained variance and therefore warrants careful interpretation. In the revised manuscript, we have clarified that the high R² reflects the inclusion of conceptually proximal and theoretically grounded predictors that collectively capture key behavioral, institutional, and contextual drivers of increasing antibiotic resistance in the study setting. We have therefore added a paragraph discussing the plausibility and limitations of the high R², emphasizing that while it indicates strong explanatory capacity within the Bangladeshi context, replication in different settings, the use of longitudinal designs, and external validation would be necessary to assess the model’s generalizability and predictive robustness. We believe this balanced discussion strengthens the manuscript by situating the statistical results within their appropriate practical and methodological boundaries.

9. Consistency between theory and empirical model

9. “The manuscript cites several theoretical frameworks (TPB, HBM, Ecological Systems Theory, Principal–Agent Theory), but the link between these theories and the exact SEM model and measured variables is sometimes implicit. Would you consider clarifying more explicitly how each theory maps onto specific constructs and hypotheses in the empirical model?

Response: We thank the reviewer for this valuable suggestion. In response, we have revised Section 2.3 to make the linkage between the theoretical frameworks and the SEM constructs explicit and transparent.

10. Generalizability and limitations

10. “The Conclusion and implications sections contain quite broad statements at global level, whereas the data come from a convenience sample in Bangladesh. Could you temper the generalizations and strengthen the Limitations section, explicitly addressing external validity and the constraints of cross-sectional, self-reported data?

Response: We thank the reviewer for this important and constructive comment. We agree that the scope of interpretation should be carefully aligned with the study design and sampling approach. In response, we have tempered the global-level generalizations in both the Conclusion and Implications sections, revising the language to clearly situate our findings within the Bangladeshi context and to avoid overextending the applicability of the results.

Reviewer #2:

The topic is clinically relevant, particularly given the global rise of antimicrobial resistance. The manuscript addresses an important public issue

All the publication criteria were satisfied, the data and the presentation is good

Great work

Response: We sincerely thank the reviewer for their positive and encouraging feedback. We are grateful for recognizing the clinical relevance of the topic, especially in the context of the global rise in antimicrobial resistance, and for highlighting its importance as a public health issue. We appreciate the reviewer’s acknowledgment that all publication criteria have been satisfied and that the data quality and presentation meet the required standards. Such constructive comments are highly motivating and reinforce the significance of our work. Thank you for your valuable time and support.

Reviewer #3:

Thanks to all authors for your great work. You choose a time-demanding topic for your research. More work and research should be done to control AMR worldwide.

I have some comments to make your work more significant. Please try to address this.

In subsection 2.2.3 (Duration of Antibiotic Use Impacts Resistance), two lines seem to be the same; you can merge/compile the lines for easy reading. The lines are, "For instance, a meta-analysis by Elshenawy [32] indicated that shorter antibiotic courses were equally effective for certain infections and reduced the likelihood of resistance development." and "For instance, Lee [34] indicated that shorter antibiotic courses were equally effective for certain infections and reduced the likelihood of resistance development."

Response: We sincerely thank the reviewer for their encouraging remarks and for highlighting this issue. We agree that the two sentences in Subsection 2.2.3 convey overlapping information and may affect readability. Accordingly, we have merged and streamlined the statements by integrating both references into a single, concise sentence that avoids repetition while preserving the contribution of each study. This revision improves clarity and ensures smoother narrative flow without altering the substantive meaning of the cited evidence. Revised: Evidence from both a meta-analysis and empirical studies indicates that shorter antibiotic courses can be equally effective for certain infections while reducing the likelihood of resistance development [32,33].

The same thing happens in '2.2.4. Healthcare Infrastructure and Diagnostic Tools Impact Resistance' and '2.2.5. Cultural Beliefs and Health-Seeking Behavior Impact Resistance.'

The lines are, "A study by Laxminarayan [12] demonstrated that countries with more substantial healthcare infrastructures and access to diagnostics reported lower levels of antibiotic resistance."; "A study by Gandra [39] demonstrated that countries with more substantial healthcare infrastructures and access to diagnostics reported lower levels of antibiotic resistance."; "For example, a study by Gautham [19] highlighted that community engagement programs in rural India successfully reduced antibiotic misuse through targeted educational campaigns." ; "A study by Gautham [20] analyzed the drivers of antibiotic provision by informal healthcare providers in rural India." Repeating lines make the readers bored during reading.

Response: We sincerely thank the reviewer for their encouraging remarks and for highlighting this issue. Revised: Previous studies have demonstrated that countries with stronger healthcare infrastructure and better access to diagnostic tools tend to report lower levels of antibiotic resistance [12,37]. Research in rural India has shown that both community engagement initiatives and the practices of informal healthcare providers play a significant role in shaping antibiotic use and misuse [19,20].

Is your collected Google Forms data from worldwide or only from Bangladesh? If it is only from Bangladesh, try to add Bangladesh in the research title.

Response: We acknowledged your observation. It was from Bangladesh only, and our revised research title is: Exploring Factors Contributing to Antibiotic Resistance: A Cross-Sectional Empirical Study in Bangladesh.

Check the citation style from top to bottom. It should be according to PLOS One policy.

Response: All references have been carefully reviewed and reformatted to strictly follow the citation style guidelines of PLOS One.

Why did you add the literature review in detail in the manuscript? You can add your relevant information in different sections as well as the introduction.

Response: In accordance with your recommendation, we have made the necessary corrections.

---

## [Decision Letter · Decision Letter 1]

22 Feb 2026

Exploring Factors Contributing to Antibiotic Resistance: A Cross-Sectional Empirical Study in Bangladesh

PONE-D-25-55672R1

Dear Dr. sharmin,

We’re pleased to inform you that your manuscript has been judged scientifically suitable for publication and will be formally accepted for publication once it meets all outstanding technical requirements.

Kind regards,

Abdullah AlSaleh

Guest Editor

PLOS One

Additional Editor Comments (optional):

Reviewers' comments:

Reviewer's Responses to Questions

**Comments to the Author**

1. If the authors have adequately addressed your comments raised in a previous round of review and you feel that this manuscript is now acceptable for publication, you may indicate that here to bypass the “Comments to the Author” section, enter your conflict of interest statement in the “Confidential to Editor” section, and submit your "Accept" recommendation.

Reviewer #3: All comments have been addressed

2. Is the manuscript technically sound, and do the data support the conclusions?

Reviewer #3: Yes

3. Has the statistical analysis been performed appropriately and rigorously? 

Reviewer #3: Yes

4. Have the authors made all data underlying the findings in their manuscript fully available?

Reviewer #3: Yes

5. Is the manuscript presented in an intelligible fashion and written in standard English?

Reviewer #3: Yes

6. Review Comments to the Author

Reviewer #3: (No Response)

7. PLOS authors have the option to publish the peer review history of their article (what does this mean?). If published, this will include your full peer review and any attached files.

Reviewer #3: **Yes:** Md. Tabeer Hossain Antor

---

## [Editor Report · Acceptance letter]

PONE-D-25-55672R1

PLOS One

Dear Dr. sharmin,

I'm pleased to inform you that your manuscript has been deemed suitable for publication in PLOS One. Congratulations! Your manuscript is now being handed over to our production team.

Kind regards,

on behalf of

Dr. Abdullah AlSaleh

Guest Editor

PLOS One